# Contemporary Strategies for Immobilizing Metallophthalocyanines for Electrochemical Transformations of Carbon Dioxide

**DOI:** 10.3390/molecules28155878

**Published:** 2023-08-04

**Authors:** Scheryn E. Lawson, Daniel B. Leznoff, Jeffrey J. Warren

**Affiliations:** Department of Chemistry, Simon Fraser University, 8888 University Drive, Burnaby, BC V5A1S6, Canada; scheryn_lawson@sfu.ca

**Keywords:** metallophthalocyanines, coordination chemistry, electrocatalysis, heterogeneous catalysis, carbon dioxide, renewable energy

## Abstract

Metallophthalocyanine (PcM) coordination complexes are well-known mediators of the electrochemical reduction of carbon dioxide (CO_2_). They have many properties that show promise for practical applications in the energy sector. Such properties include synthetic flexibility, a high stability, and good efficiencies for the reduction of CO_2_ to useful feedstocks, such as carbon monoxide (CO). One of the ongoing challenges that needs to be met is the incorporation of PcM into the heterogeneous materials that are used in a great many CO_2_-reduction devices. Much progress has been made in the last decade and there are now several promising approaches to incorporate PcM into a range of materials, from simple carbon-adsorbed preparations to extended polymer networks. These approaches all have important advantages and drawbacks. In addition, investigations have led to new proposals regarding CO_2_ reduction catalytic cycles and other operational features that are crucial to function. Here, we describe developments in the immobilization of PcM CO_2_ reduction catalysts in the last decade (2013 to 2023) and propose promising avenues and strategies for future research.

## 1. Introduction

Metallophthalocyanine (PcM) complexes are a vast class of compounds that have been known since the early 20th century; for some historical reviews, see [1,2]. The core structure of PcM is set out in Figure 1. Perhaps their most readily identifiable characteristic is their vivid blue and green colors. The high thermal stability of many PcMs is also a noteworthy feature. Together, the two aforementioned properties of PcMs have led to their widespread use as colorants [3]. However, they also have garnered attention in a surprisingly large number of other applications, from energy [4,5] to biomedical applications [6,7] to catalysis [8]; however, those represent only a few examples and accompanying references. Indeed, research efforts continue to grow, with a query of “phthalocyanine” using any common search tool (SciFinder, Web of Knowledge, Google Scholar, etc.) garnering hundreds of thousands of hits in both the primary and patent literature. Of special concern in this review is the applications of PcMs as mediators of the electrochemical transformation of carbon dioxide (CO_2_).

The goal of this review is to outline recent results with respect to heterogeneous CO_2_ reduction mediated by different PcM complexes and materials. We choose not to make an extensive list of performance metrics because there is not yet an accepted standard by which to benchmark disparate systems. There are examples of other recent reviews that tabulate such metrics [5,9,10,11], but doing so is a known challenge in related areas of electrocatalysis [12,13,14,15,16]. Instead, we try to focus on recent approaches to making molecules and materials that have favorable CO_2_ reduction kinetics and/or overpotentials. In this vein, we also emphasize areas where research has lagged or can be further expanded. 

The development of systems that can upgrade CO_2_ to fuels gained momentum during the gas crisis of the 1970′s and was reinvigorated by our present need to address climate change. There is ongoing discussion regarding the most desirable product of CO_2_ upgrading [17,18,19] and similar discussion about the technology needed to achieve CO_2_ capture and conversion on a meaningful scale [20,21,22]. Molecular electrocatalysts have received attention due to their intrinsic synthetic modularity, which enables detailed studies of structure–function relationships (see, for example, [23,24,25] and references therein). We note that there is a corresponding body of work that focuses on purely inorganic materials, but that is beyond the scope of this review. 

Cobalt(II) and nickel(II) phthalocyanines were among some of the earliest molecules reported that can mediate the electrochemical reduction of CO_2_ to CO [26,27]. Another early report expanded on those initial results and showed more decisively that PcCo is a very good CO_2_ reduction catalyst [28]. Further improvements are possible with addition of electron withdrawing groups [29]. In the 50 years since those reports, there have been investigations of many different metals (M) in PcM complexes that can reduce CO_2_ to C_1_ products and can sometimes effect proton reduction to make H_2_ [30,31,32]. These works reinforced the idea that PcCo and PcNi are attractive catalysts, but they are by no means the only PcMs that can mediate reduction of CO_2_. Herein, we primarily focus on recent (ca. 2013–2023) developments in immobilized PcM systems that afford promising CO_2_ reduction activity or useful lessons in the design or new systems and molecules.

The following sections each give a short overview and some historical context for different heterogeneous preparations of PcM. The initial section provides some background and different viewpoints regarding homogeneous and heterogenous catalysis for PcM. That section is followed by a short discussion of CO_2_ reduction mechanisms, with emphasis on PcCo, where there is a great deal of work. The following sections describe different approaches to the generation of PcM materials for CO_2_ reduction. We also take the opportunity to highlight areas where we think that there are promising areas open for exploration. We have tried to categorize the different approaches to preparing PcM-modified materials into broad categories. We emphasize that there is conceptual and practical overlap, and it is exciting to see where different approaches to designing and investigation materials for CO_2_ reduction agree and contrast.

## 2. Heterogeneous versus Homogeneous Catalysis

Homogeneous electrochemistry and related solution experiments are common starting points for investigations of molecular catalysts [14,16,33,34,35]. While there exist many tools for benchmarking electrocatalysis in a solution, there also are a dearth of examples of homogeneous CO_2_ reduction using PcM. One early example explored the electrochemistry of tetra-β-sulfonato-PcM, where M = Fe, Co, Ni, or Cu (**1** in Figure 2) [27]. The Co and Ni complexes showed increased CV currents in the presence of CO_2_, which was suggested to be evidence for CO_2_, although the products were not reported. The corresponding Fe and Cu complexes did not show any increases in current in the presence of CO_2_. For Co and Ni, the authors concluded that a M-CO_2_ complex formed “on the electrode”, but they did not distinguish between homogenous and heterogeneous catalysis. Likewise, the products of CO_2_ reduction were not described. Surprisingly, there appears to have been no follow-up investigations of the homogenous CO_2_ reduction properties of sulfonated PcM. 

In general, there are few reports of the homogeneous CO_2_ reduction electrocatalysis of PcM complexes. Their low solubility and tendency for aggregation are underlying factors. Distinguishing between homogenous and heterogeneous pathways for electrocatalysis by large and (often) hydrophobic coordination complexes is a known challenge, as was shown using cyclic voltammetry and hydrodynamic electrochemistry with iron-porphyrin dioxygen reduction catalysts [36]; this is further exacerbated for ring-unsubstituted PcM complexes, which often have a low solubility and a high tendency to aggregate. A related investigation for CO_2_ reduction using both an anionic and a cationic PcCo complex was recently reported (**2** and **3** in Figure 2) [37]. Compound **2** is an especially interesting example as it can be neutral or charged, depending on solution pH. In that work, the effect of catalyst adsorption to carbon electrodes was explored for CO_2_-to-CO conversion and for CO-to-CH_3_OH conversion. Using a series of electrochemical experiments with soaking/washing steps to remove weakly adsorbed catalysts, it was shown that the reductive electrocatalysis for both CO_2_ and CO reduction is dominated by heterogenous reactions of adsorbed molecules regardless of the charge on the ring-substituent groups. The Faradaic efficiencies for both CO_2_ and CO reduction were slightly higher for the anionic (i.e., deprotonated **2**, Figure 2) PcCo catalyst. For CO to CH_3_OH conversion, the Faradic efficiency was 10% for compound **2** and 5% for compound **3** and for CO_2_ to CO conversion, the Faradic efficiency was 90% for compound **2** and 75% for compound **3**. The cyclic voltammetry data suggest that both catalysts are active at the same potentials [37]. Further investigation into how PcM ring-substituent groups affect catalysis, aggregation, and adsorption would be worthwhile. For example, the structure–function relationships for non-aggregating PcM derivatives have been extensively discussed for photochemistry, with respect to biomedicine [38] and solar energy [39].

There exists a contrast between the development of metalloporphyrin and PcM CO_2_ reduction catalysts. This contrast feeds into the discussion of mechanism in the following section. Whereas investigations into metalloporphyrin electrocatalysis have relied on benchmarking as homogeneous catalysts (e.g., [23,40] for reviews), investigations into PcM have involved a wide variety of heterogeneous preparations, perhaps in part due to the relative insolubility and high thermal stability of ring-unsubstituted PcM complexes. We think that understanding the homogeneous chemistry of soluble PcM derivatives is of great interest and would inspire new thinking about how to design PcM complexes in ways that can augment CO_2_ reduction kinetics, energetic efficiency, and/or product selectively. Ultimately, an improved understanding of PcM CO_2_ reduction mechanisms and reaction bottlenecks is still sought. 

## 3. Selectivity and Mechanisms of CO_2_ Reduction

The identity of the metal (M) in PcM affects CO_2_ reduction properties, in particular influencing the selectivity for a given product. The range of CO_2_ reduction products from different metals was recently summarized [32] based on foundational work using PcM in gas-diffusion electrodes (GDEs) [30,31]. For a recent review on GDEs and their applications in CO_2_ conversion, the reader is pointed to reference [41]. PcCo, PcNi, and PcPd are reported to give predominantly CO as the CO_2_ reduction product. Early transition metals afforded primarily H_2_ (i.e., proton reduction) and main group metals, such as lead and tin reducing CO_2_ to formic acid, with some degree of proton reduction to H_2_. Interestingly, PcTi-based species and PcCu gave a mixture of products that included CH_4_ as a large fraction. Here, we will focus our mechanistic discussion on CO since this is perhaps the simplest product, requiring the addition of 2H^+^ and 2e^−^ to CO_2_. In general, CO is thought to be produced first and further reduction leads to other C_1_ products (e.g., CH_3_OH or CH_4_) [42]. Our understanding of the features of molecules or of materials that give rise to different selectivity remains limited. The range of metals was systematically described over 30 years ago [30,31] and, given advances in catalyst design and electrochemical analyses, we think that a wider range of M in PcM complexes deserves attention.

The mechanisms by which different first row transition metal PcM complexes produce CO or HCOOH has been investigated using density functional theory (DFT) [43]. Those calculations support the idea that PcCo is a good catalyst for CO_2_-to-CO conversion using the sequence of reactions set out in Figure 3. Unfortunately, the oxidation state of the complexes in the DFT were not stated explicitly. Nonetheless, the calculations suggest that the initial CO_2_ bonding step (step i, Figure 3) has about the same energetics for PcM, where M = Mn, Fe, Co, Ni, or Cu. The calculations also indicate, for the case where M = Co, that the energy associated with the two protonation reactions of the [PcCo-CO_2_] intermediate was modest (steps ii and iii, Figure 3). Likewise, the binding energy of CO to the cobalt ion was small enough to allow facile CO release (step iv). In the end, PcCo was deemed a superior catalyst because the barrier for [PcM-COOH] formation is balanced with the ability of the [PcM-CO] intermediate to release CO and re-enter the catalytic cycle (following step iv in Figure 3). Complementary experiments supported the results of those calculations [43].

Experimentally, the mechanism of PcCo-catalyzed CO_2_ reduction is the most widely discussed, starting even with some of the early reports of its activity (Figure 4) [44]. A commonly accepted mechanism invokes formation of a [H•PcCo] intermediate (intermediate **c** in Figure 4) following the first reduction of the PcCo(II) complex. We note that the location of the proton is not conclusively known, but it is thought to reside on the ligand, not the Co ion. A subsequent reduction of [H•PcCo] affords [H•PcCo]^−^_,_ which can bind and activate CO_2_ (blue and green colored paths, Figure 4) or provide a route for proton reduction to give H_2_ (red colored path, Figure 4). In instances of low proton activity and alkaline conditions, it may be possible for the singly reduced [PcCo]^−^ to activate CO_2_ directly [45,46].

As pointed out for two very-different PcCo preparations (polymer encapsulation [47] versus dispersion on carbon paper [45]), there are numerous factors that can affect mechanistic studies, from local environment effects to metal–ligand interactions to aggregation/dispersion. One example of this is the contrast between an early study of PcCo and 1,4,8,11,15,18,22,25-octacyano-PcCo, and poly(4-vinylpyridine)-encapsulated PcCo (PVP•PcCo). In the former case, experiments suggested that the rate-limiting step for PcCo was a second reduction step to access a formally Co(0) complex, but the introduction of the cyano groups shifted the rate limiting step to CO_2_ binding [29]. In the PVP•PcCo case, the rate limiting step is the binding of CO_2_ to the protonated anionic intermediate (intermediate **d** in Figure 4). While the mechanisms are similar, the differences underscore the importance of considering structure–function relationships in catalyst system designs. In addition, the presence of different functional groups also can affect mechanisms and kinetics [29,44,48], most likely via changes to formal potentials in a Hammett-like trend (see Section 5 for a related example). Finally, we note that there is a lack of experimental mechanistic data on other PcM complexes, and it would seem that this is an area ripe for exploration.

The above mechanistic proposals are supported and informed by different in situ experiments. Such techniques continue to advance and will remain important tools for probing mechanisms. For example, the DFT calculations in reference [43] were supported by in situ X-ray absorption spectroscopy (XAS) measurements. Using similar systems where PcM were deposited on conducting carbon, those results were later corroborated by DFT, in situ powder X-ray diffraction, X-ray absorption near edge spectroscopy (XANES), and Extended X-ray absorption fine structure (EXAFS) [49]. PcCo again showed the highest selectivity for CO_2_-to-CO conversion and the spectroscopic measurements confirmed that the N_4_-ligated metal ions remained intact during operation. For PcCo, reaction intermediates (proposed as [PcCo-CO_2_]^−^) also have been detected in PcCo monolayers on gold using in situ scanning tunneling microscopy (STM) experiments [50]. In the case of polymer-encapsulated PcCo, XANES was used to show that a 5-coordinate [N_5_Co] site is the dominant species, but that the population is dependent on pH. In contrast to PcCo, in situ XAS measurements on carbon nanotube-supported PcCu showed that Cu nanoparticles reversibly form in a potential-dependent manner and the results suggest that the nanoparticles are the catalytically active form of Cu [51]. While this is a small survey of in situ studies, a blend of electrochemical modeling, DFT, and in situ spectroscopy will be crucial in further understanding the mechanisms of a new PcM CO_2_ reduction catalyst.

## 4. Direct Electrode Adsorption/Deposition and Catalyst Inks

The earliest reports of PcM electrocatalysis employed simple drop-casting to adhere the complexes to conductive surfaces (commonly carbon electrode materials) [26,28] and this approach remains common in the early stage benchmarking of PcM catalysts. Drop-casting, and related methods that are described below, leverage the inherently low solubility of ring-unsubstituted PcM in common organic solvents to promote interactions with a conductive substrate. In the presence of axial ligands that come from donor solvents such as DMF (see below), PcMs have some solubility that does facilitate solution processing. However, we now know that the natural aggregation of phthalocyanines [52] is a significant issue that limits catalytic rates, likely by limiting the accessibility of the active sites. More highly dispersed PcCo on carbon paper shows higher turnover frequencies (≥10^2^ s^−1^) for CO_2_-to-CO conversion, though off-target proton reduction contributes more to the current response as the catalyst becomes more highly dispersed [45]. The lowest loadings also show Tafel slopes close to 120 mV per decade, which suggests a rate-limiting electron transfer step. In contrast, higher loadings (i.e., more aggregation) have large Tafel slopes, suggestive of transport limitations. 

One common way to improve the solution processability and dispersity of PcM complexes is through the addition of bulky alkyl-containing groups to the α or to the β positions (Figure 5), often as alkoxy- or phenoxy-substituents. In general, substitution at the β positions has less of an effect on the electronic structure of the macrocycle than substitution at the α positions [53]. Although a full treatment of PcM syntheses is beyond the scope of this review, we note that the number of known phthalocyanine structures is enormous. Most of the work described in this review focuses on the more inexpensive or synthetically accessible compounds, which are likely to be essential for large-scale application. In general, those PcMs that are both inexpensive and straightforward to prepare usually include those that are symmetrically octa-substituted at the α or β positions (or both) and those that are tetrasubstituted at either position (usually generating a mixture of isomers); examples of such compounds can be found in Figure 5.

In one recent example featuring the PcM β positions, substitution of all eight β sites with linear -OC_8_H_17_ chains (**4** in Figure 5) was shown to improve the effective turnover frequency (TOF) of CO_2_ electrocatalytic reduction by PcCo by a factor of 2–3 with respect to unsubstituted PcCo. In this example, both complexes were deposited on chemically converted graphene from DMF or THF solution and then drop-cast on carbon paper [54]. The modest steric and electronic effect of substitution at the β position allows for a more clear comparison between the two complexes with very different sizes and aggregation profiles.

Like unsubstituted PcCo, octa-α-substituted PcM are known CO_2_ reduction catalysts [29,48], but only recently were the properties of different structures and preparations explored. An investigation using operando Raman spectroscopy on carbon-deposited PcCo and the modified version 1,4,8,11,15,18,22,25-octaethoxyphthalocyanine cobalt(II) (α-EtO_8_PcCo, **5** in Figure 5) showed that higher catalyst loadings led to a decrease in the fraction of Co(II) sites that were reduced under operating conditions [55]. Importantly, the authors highlight the challenges in the reproducibility of molecular catalysts deposited onto solid supports. Here, the deposition was onto carbon from a suspension in pyridine. In addition to the effect of aggregation, the authors demonstrated that the incorporation of the electron-donating alkoxy groups increases electron density at the Co site, which is proposed to be a driver of increased CO formation rates. Turnover frequencies of about 10^2^ s^−1^ are possible and Tafel slopes near 120 mV per decade are consistent with rate limiting electron transfer. 

In work from our group, we explored the relationship between the degree of aggregation and the type of substitution at the α position [56]. In simple drop-cast preparations of **6**–**8** (Figure 5), we demonstrated that bulkier alkoxy-chains do not necessarily lead to better dispersion and, therefore, more facile access to active sites. In fact, for **8**, noticeable solution aggregation was observed using UV-vis spectroscopy. The complex with *sec*-butoxy groups (**7** in Figure 5) showed the greatest peak current at the lowest loadings (deposited at ca. 10^−11^ mol cm^−1^). The additional steric profile of the *sec*-butoxy groups appears to allow better access to Co sites, promote more even dispersion, and the groups are sufficiently small as to not introduce another mode of aggregation, such as interactions between the aliphatic substituents.

Another recent report demonstrates that the addition of pyridine groups to four of the β-positions (denoted –OPy in **9**, Figure 5) can diminish CO_2_ reduction overpotentials and give turnover frequencies of ca. 7 s^−1^ in dip-cast carbon paper assemblies [57]. In contrast to the pyridine-containing materials described below, the authors do not associate the improvement in CO_2_ reduction activity with any degree of ligation of the Co ion by the pyridine groups. Such Co-pyridine interactions are certainly possible in the solutions used for catalyst deposition (trifluoroethanol or DMF), and the pyridines are likely not protonated under operational conditions at pH 6.8, but the authors suggest that pyridine plays a purely electronic role in tuning the reduction potentials of the Co site. A series of DFT calculations show that **9** has a stronger electron affinity than the parent PcCo and is, therefore, more easily reduced to the double-reduced species, which is the intermediate that binds and activates CO_2_ (similar to intermediate **d** in Figure 4). In addition, the DFT analyses showed that the CO_2_ binding energies are about the same for the pyridine-modified complex and for the unsubstituted PcCo. 

Many of the studies described in this section and in the next involve the production of what are broadly termed “catalyst inks.” The inks usually consist of a catalyst mixed with a conductive carbon support (conductive carbon materials such as Ketjen black, carbon nanotubes, etc.) and dispersed in an ionomer, such as Nafion. Such catalyst preparations are widely used in fuel cells and the challenges associated with the multiple interfaces involved in such systems are appreciated [58]. A partial list of such operationally important interactions includes the interactions of catalysts with solid particulate supports, the interparticle interactions, the particle ionomer interactions, and the solvent–ionomer interactions. Work in this area is ongoing for CO_2_ electrolyzers and many of the same challenges persist. Testing in lab-scale electrolyzers clearly has an important place in testing novel PcM materials (e.g., a highly active PcCo cation [59]), but considering dispersity in inks is also critical. In addition, many of those works underscore the importance of catalyst dispersion. It remains a challenge in PcM-mediated catalysts to maximize the number of active sites through dispersity. 

## 5. PcM Materials Supported on Carbon Nanotubes

The above section focused on materials where PcM was adsorbed to macroscopic carbon-based electrode materials. However, improvements to catalysis are also possible when PcM (especially PcCo) is deposited on carbon nanotubes (CNTs). In terms of redox catalysis, the hypothesis is that the highly conductive framework of CNTs enhances the flow of electrons to catalyst sites. There have been many other reports of CNT-immobilized PcM complexes in a range of other applications, but these are beyond the scope of this review. These, and related, CNT scaffolds afford a route to disperse coordination complexes in a way that can enhance their physical properties, including redox catalysis.

Two reports of PcCo immobilized on CNTs from different groups appeared nearly simultaneously in 2017, though the approaches used to generate the materials were distinct [46,60]. In reference [46], the PcCos were prepared via microwave-assisted condensation directly onto CNT substrates using 1,2,4,5,-tetracyanobenzene and CoCl_2_. The result of the condensation is a core-sheath arrangement with the CNT at the core and a crosslinked PcCo material as the sheath; an example of a crosslinked subunit is shown in Figure 6, i.e., **10**. The thickness of the PcCo material was ca. 5 nm and the weight percent of cobalt in the materials was about 2.6%. The other material [60] was made from DMF suspensions of PcCo and CNTs, where the PcCo is anchored via non-covalent (likely π–π) interactions. In this case, the cobalt weight percent was about 0.26%. Of special note is that both systems achieve similar overpotentials and turnover frequencies (ca. 1–5 s^−1^). This observation highlights the importance of per-site activity; the crosslinked PcCo-CNT system has a factor of 10 more Co to achieve the same overall activity. 

A modified PcNi complex immobilized on CNTs was recently reported (**11** in Figure 6) [61]. The material can be formed in situ from CNTs, the 2,3,9,10,16,17,23,24-octahydroxyphthalocyanine Ni(II) complex, and 2,3,5,6-tetrafluoro-1,4-dicyanobenzene, and a model complex can be prepared by omitting the CNTs. The CNT assembly was reported to have nearly quantitative Faradaic efficiency for CO_2_-to-CO conversion over a potential range similar to PcCo-based materials. Kinetics parameters were not reported, but a mass percentage of 22.1% for **11** (Figure 6) had the best response based on maximum currents. That loading corresponds to 1.5% mass percent Ni(II). While mechanistic data are still limited for complexes other than PcCo, DFT calculations suggest that **11** (Figure 6) has a lower barrier to reduction and protonation of a [PcNi-CO_2_]^−^ intermediate than for the octa-hydroxy complex or for the parent PcNi. Like PcCo, PcNi has good yields of CO from CO_2_ reduction [30,31,32] and additional research into PcNi CO_2_ reduction chemistry could be promising.

The dispersed PcCo•CNT composite materials [60] described above were further investigated and it was demonstrated that the materials could mediate the conversion of CO_2_ to CH_3_OH in a potential-dependent manner [42]. In this process, CO is a required intermediate and the overall conversion of CO_2_ was termed “domino electroreduction”. The CNT composite materials made from PcCo showed modest stability, which was attributed to reduction-linked degradation of the ligand, but no Co-containing nanoparticles were observed. The corresponding material prepared from the mixture of isomers of tetra-β-amino PcCo (i.e., one NH_2_ group at one β position on each of the four benzenoid rings) showed improved stability. The electron-donating NH_2_ groups make the complex more difficult to reduce (i.e., lower the PcCo reduction potentials), which could be the origin of the greater stability at the potentials required for CO_2_ reduction. Likewise, an increase in electron density at the Co ion could improve CO_2_ or CO activation. A remarkable 30% Faradaic efficiency for CH_3_OH was reported. To the best of our knowledge, PcCo is still the only molecular catalyst to give such good yields of CH_3_OH from CO reduction. 

Other approaches to immobilizing PcCo onto CNTs involve initial modification of the CNTs with functional groups that can act as ligands to the open axial sites in PcM complexes (Figure 7). Such additional ligands can affect the properties of the metal site, as described below in the cases of PcCo encapsulated in ligating polymers. Addition of pyridine to CNTs and addition of PcCo gives rise to materials that support CO_2_-to-CO conversion [62]. The pyridine groups aid in both dispersing the PcCo and in tuning the electronic properties of the Co site. In comparison to other dispersed PcCo [45], the reported rate constants are about a factor of three faster. At higher PcCo loadings, the rate constants become independent of deposition of CNTs or macroscopic substrates, reinforcing the idea that aggregation and occlusion of active sites is a crucial consideration of PcM preparations. CNTS also can be modified to include -COOH, OH, or NH_2_ groups along their backbone (Figure 7) [63]. All three of those groups promote better dispersity and therefore better CO_2_ reduction activity, but the NH_2_-modified CNTs show the overall best kinetics. Interestingly, the turnover frequencies for the NH_2_- and pyridine-modified CNTs are about the same (ca. 30 s^−1^). In both of the above examples, the Tafel slopes are near 120 mV per decade, suggesting that an electron transfer is rate limiting and, in contrast to higher loadings, transport limitations are absent.

Cross-linked PcCo 2D materials, where the properties of catalyst active sites are modified by functional groups on the Pc, also have been prepared using carbon nanotubes as a solid conducting support [64]. The functional groups surveyed were remote from the Co site (Figure 8), but they did influence the apparent Co(II)/Co(I) reduction potentials. The potentials showed linear correlations with Hammett parameters, and they shifted by about 200 mV between the biggest electron-donating group (NH_2_) and the biggest electron-withdrawing (CO_2_H). Likewise, the more electron-withdrawing groups gave rise to ca. 10% larger turnover frequencies than the electron donating groups. All of the materials prepared were active for CO_2_ reduction, where CO was the main product (≥80% Faradaic efficiency); proton reduction accounted for the remainder of the activity. The materials all had TOF values of approx. 3 s^−1^, with a weak correlation with Hammett parameters. The authors also note that just 14–18% of the Co sites are active. In many ways, these materials are related to the extended network materials described in the following section.

## 6. Network Materials Built with PcM

Covalent organic frameworks (COFs) are a class of two-dimensional materials where subunits are connected via covalent bonds; the basics of COF chemistry are reviewed elsewhere [65]. COFs share some similarities to metal-organic frameworks (MOFs), where the latter is distinguished by their assembly using metal–ligand coordinate bonding and thus includes the presence of heavier elements/metal centers as components/linkers of the network. The preparation of covalent- and metal-organic frameworks based on metalloporphyrins [66,67] spurred research into the corresponding PcM-based materials. Some of the first examples were not designed as catalysts, but were of interest as conductors/semiconductors (e.g., [68,69]). One of the first examples of a CO_2_ reduction system derived materials from crosslinked 2,3,9,10,16,17,23,24-octaaminophthalocyanine Co(II) and 4,5,9,10-pyrenediquinone (**12** in Figure 9) [70]. The material displayed near perfect (96%) Faradaic efficiency for conversion of CO_2_ to CO at rates of approx. 3 s^−1^. The Tafel slopes suggest that electron transfer was rate-limiting, not mass transport. Similarly, two-dimensional sheets of polyimide-linked PcCo (**13** in Figure 9) show good Faradaic efficiency for CO production in an overpotential-dependent manner (70–90+%) and have turnover frequencies of ≥2 s^−1^ [71]. Those materials also have chemical stability in concentrated HCl and thermal stability to ca. 300 °C.

PcM-derived conjugated metal-organic frameworks (c-MOF) are materials that have also recently emerged. Perhaps the first example was the description of materials designed for the detection of potentially hazardous gases [72]. These materials featured PcNi subunits linked via [NiO_4_] subunits, where the O comes from a catecholate (Figure 10, where M_A_ = M_B_). The square planar Ni-catecholate sites are best described as Ni(II) ions ligated by *ortho*-semiquinones or quinones, depending of the structure of the aromatic system [72]. A series of structurally related heterobimetallic systems consisting of Cu and Zn sites was later shown to be competent for CO_2_ reduction [73]. The system where M_A_ = Cu(II) and M_B_ = Zn had nearly 90% Faradaic efficiency for CO production and ca. 10% afforded H_2_ production. The turnover frequency for CO production was 0.4 s^−1^ and the material withstood testing for over 10 h. The other materials, either homometallic Cu(II), Zn(II), or the heterometallic mixture with Zn(II) at the PcM sites, were less effective CO_2_ reduction catalysts.

The aforementioned [72] materials featuring PcNi-linked via square-planar nickel(II) sites (M_A_ = M_B_ = Ni(II), Figure 10) were later used for the reduction of CO_2_ [74]. DFT calculations of the CO_2_-reducing system suggest that the Ni(II)-catecholate sites are not directly involved in CO_2_ activation; i.e., the PcNi sites are responsible for CO_2_ activation [74]. The Faradaic efficiency is about 98% for conversion of CO_2_ to CO and the turnover frequency was about 0.7 s^−1^. 

Two dioxin-linked COF materials were developed following the synthetic method in Figure 11 [75]. Here, a base-catalyzed nucleophilic aromatic substitution of perfluorinated PcCu and hydroxy-substituted PcCo gave rise to materials with variable pore sizes that were dictated by the size of the aromatic groups on PcCo. The best Faradaic efficiencies for the *n* = 1 and *n* = 2 materials were 91% and 97%, respectively. The corresponding turnover frequencies were 1.3 s^−1^ (*n* = 1) and 2.9 s^−1^ (*n* = 2) and the Tafel slopes near 120 mV per decade indicate that electron transfer is the rate limiting step in catalysis. In general, the complex with the larger pores and more conjugated π-system (i.e., *n* = 2, a naphthalocyanine core), showed better CO_2_ reduction characteristics.

The concept of catalytic COF and MOF materials is attractive, but some general features appear from this survey of materials. First, the overall ease of preparation and stability of the materials appears to be excellent, at least in lab-scale tests. However, the number of active sites also appears to be modest, limiting turnover frequencies to less than 10 s^−1^, and in some cases, less than 1 s^−1^. Based on Tafel slopes, the modest rate constants are not due to slow mass transport, but rather due to slow electron transfer. Most of the PcM-based materials feature arrangements of 2D sheets, which is due to the inherently flat Pc framework. As such, the inter-layer electron flow is likely much slower than electron flow along the conjugated system. For example, interfacial electron transfer rate constants can be as much as a factor of 10^5^ faster on pure edge-plane graphite versus pure basal plane graphite [76]. Thus, it may be of interest to test some of these materials in other preparations to probe how their performance and stability is related to the 2D stacked structures.

Many of the above examples probe materials with different pore sizes, which are dictated by the nature of the linker group (e.g., the diamine in compound **13**, Figure 9). Pores have a strong influence on mass transport and are therefore important in these catalytic materials. Thus, the pore size can have an impact on the performance of a given material, but there are not yet clear rules with respect to the design of 2D PcM-based materials. For example, the material shown in Figure 11 has a higher Faradaic efficiency at lower overpotentials when *n* = 2 versus *n* = 1 (i.e., the larger pores have better performance). The overall better performance was proposed to be due to both pore size and the conductivity of the material. In contrast, for compound **13**, the smaller pore size complexes (*n* = 1 for the amine linker) has somewhat better performance than the compound with an *n* = 2 linker, at least based on current density for CO_2_ to CO conversion [71]. However, the difference in observed turnover frequency is small: 1.9 s^−1^ for *n* = 2 and 2.2 s^−1^ for *n* = 1. Investigations of pore sizes and the interlayer spacing of 2D PcM-based COFs are important areas for continued research. 

Finally, we think that PcM-based network materials, and the highly dispersed preparations discussed in Section 4 and Section 5, are a connection point to emerging single-site catalytic materials. In fact, PcM have been used as models for the CO_2_ reduction properties of such materials, as described above [43]. Research into single-site catalysts is an active area and a complete evaluation is not possible here (for a recent review, see [77]), but we will provide a handful of examples. Computation has been used to survey a range of N_4_-ligated metal ions in nitrogen-doped graphene [78] and graphyne [79] where the structures are reminiscent of PcM. In both cases, first-row transition metals are predicted to have favorable CO_2_-to-CO conversion properties. In analogy to some of the above studies on PcM, single-site nickel materials are robust (over 100 h operation) and can produce CO with turnover frequencies of about 4 s^−1^ [80]. Single cobalt ions in nitrogen-containing carbon nanospheres extend the idea of single-site macroscopic materials into smaller, nanoscale materials, though the turnover frequencies remain comparatively small (ca. 0.1 s^−1^). In a similar vein, ligand-supported metal nanoclusters are competent CO_2_ reduction electrocatalysts and share some similarities to PcM-based materials and single site catalysts [81]. In the long term, continuing to build conceptual connections between molecules and materials is important and PcM molecules are a great starting point. 

## 7. Polymer Incorporation

The previous sections focused on materials where PcM was non-specifically incorporated into solid supports or made up a structural portion of the material (i.e., COFs). Another class of PcM-derived materials includes polymers that are covalently modified with PcM, either as an added component post-polymerization or as a monomer. For example, one early example showed how polythiophene-based PcM could be prepared and polymerized (e.g., from **14** in Figure 12), but their activity toward CO_2_ reduction was not probed [82,83,84]. Recently, this idea was re-visited in the context of CO_2_ reduction using **15** (Figure 12) [85]. Compound **15** is readily electropolymerized on carbon paper or indium tin oxide-coated glass and the resulting polymers contain the expected ratios of Co and S derived from the molecular structures. A range of conditions and operating times were assayed; under ideal conditions (2 h electrolysis) the Faradaic efficiency was 94% and the turnover frequency was 0.3 s^−1^. Importantly, a 20 h electrolysis showed a ca. 30% decrease in turnover frequency (to 0.2 s^−1^) and about a 25% decrease in Faradaic efficiency (to 72%). As noted above for MOF and COF materials, the stability of this system is noteworthy, but the slow turnover is problematic and may be due to a large number of inactive sites. It may be advantageous to co-polymerize thiophene and **15** as a way to increase the dispersity of PcCo.

Compound **16** (Figure 12) has previously been prepared in a material where it was proposed to crosslink strands of polypyrrole [86]. In this example, pyrrole was oxidatively (chemically) polymerized in the presence of **16** [86,87]. The Faradaic efficiency was reported to be better than 93% and the turnover frequency was 0.4 to 0.6 s^−1^, depending on the potential and duration of electrolysis. The authors propose that the nitrogen-rich polypyrrole polymer can help to accumulate CO_2_ near the catalytically active PcCo sites. The composites showed ≥90% Faradaic efficiency down to 18% CO_2_ in argon (the lowest concentration tested). We note that the turnover frequency values were similar to the polythiophene polymers above, which could suggest that polymer synthesis in the presence of PcCo limits access to active sites. As such, the modification of existing polymers with PcM, as described below, could be favorable.

Given the low solubility of ring-unsubstituted PcM in many common organic solvents, one strategy to improve dispersion/accessibility of active sites is to disperse the PcM catalysts in polymers. To that end, one promising candidate is poly (4-vinylpyridine), where the pyridine was initially proposed to aid solubility by acting as a ligand to the metal center and to promote catalysis by improving proton transfer reactions [44,88]. However, at the time, these proposals were not verified and a more comprehensive study using a 5-coorindate PcCo-pyridine model and different polymers confirmed the synergistic effect that poly (4-vinyl)pyridine played on CO_2_ reduction [89]. In situ X-ray absorption spectroscopy experiments confirmed that PcCo is in a 5-coordinate form and that lowering the pH (thereby protonating the pyridine groups) leads to loss of coordination [90]. Ultimately, it is clear that both metal-ligation and proton transfer are important facets of the CO_2_ reduction chemistry of polymer-encapsulated PcM (M = Co in this case). An ideal PcCo-PVP system has over 90% Faradaic efficiency and turnover frequencies approaching 6 s^−1^; a nice comparison of different preparations is given in reference [91]. An important open question revolves around other second-sphere interactions of pyridine. There are known examples of the impact of nitrogen-containing functional groups in the second sphere of coordination complex catalysts that mediate CO_2_ reduction, e.g., [92,93,94,95,96,97].

## 8. Conclusions and Outlook

Metallophthalocyanines will continue to be chemically important coordination complexes in a range of applications. While their use as dyes was firmly established decades ago, their applications continue to grow, in particular in electrocatalysis. One common thread that emerged from this survey was the general stability of the PcM core for electrocatalytic applications. Overall, the past decade has seen expanded interest in re-visiting PcM CO_2_ reduction chemistry that emerged over 40 years ago. Work is ongoing by electrochemists, materials scientists, and engineers. The robust PcM scaffold is promising and many of the core principles in the design of functional materials are known. The ability to disperse metal active sites in materials is of great use and the field remains open to a range of explorations, some of which we note below.

There remain challenges in the use of PcM as mediators of electrochemical CO_2_ reduction. First, it is now clear that the dispersity of molecular catalysts is a crucial design consideration. The development of a strategy to combine the stability of metal-organic or covalent organic frameworks with the higher per-site activity of PcM deposited on carbon materials, or in polymers, is desirable. One important metric is turnover frequency; highly dispersed preparations can turnover at rates surpassing 10^2^ s^−1^, where most other materials turnover at rates of less than 10 s^−1^. A detailed study of how operational conditions affect CO_2_ reduction kinetics also would be useful. Next, revisiting the chemistry of other PcM complexes using modern material preparations could yield novel chemistry. Such explorations should include detailed mechanistic work, especially for metals other than cobalt. Understanding mechanisms using solution electrochemistry and heterogeneous methods (e.g., analysis of Tafel slopes) will continue to be essential. As noted above, the analysis of operational conditions is also desired with respect to the mechanism. Finally, there remains relatively little known about how the second coordination sphere of PcM coordination complexes affects their catalytic activity or their ability to be incorporated into functional materials. Advances in all of these areas can be anticipated to further improve the viability of PcM electrocatalysts to convert CO_2_ into valuable fuels and other chemical feedstocks.

## Figures and Tables

**Figure 1 molecules-28-05878-f001:**
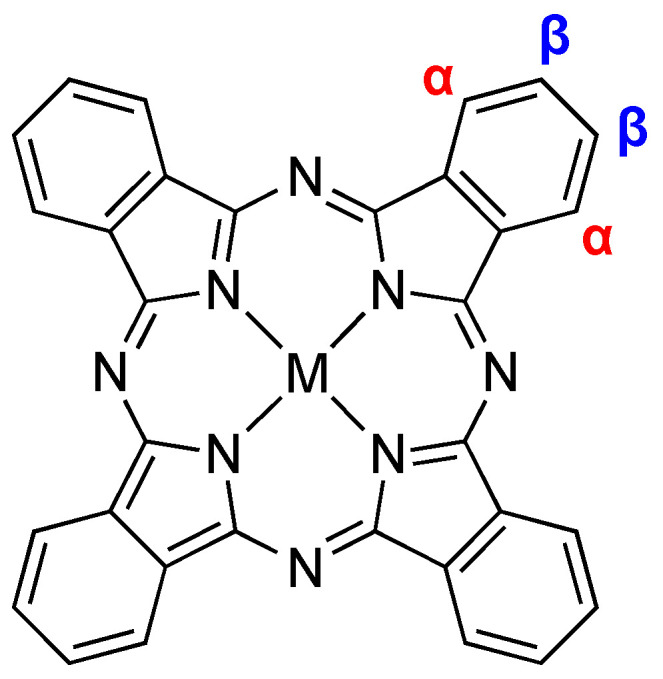
Basic structure of M(II) phthalocyanine complexes. The non-peripheral (α) and peripheral (β) ring-substitution sites are labelled.

**Figure 2 molecules-28-05878-f002:**
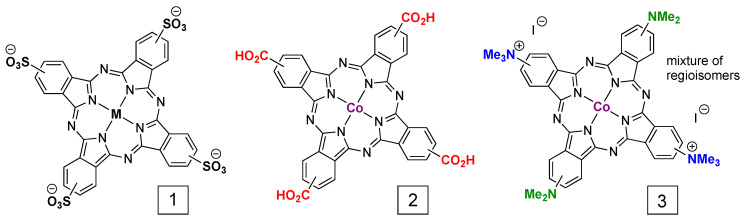
Structures of metallophthalocyanine complexes used to investigate homogeneous electrochemical CO_2_ reduction. Compound **3** includes a mixture of regioisomers.

**Figure 3 molecules-28-05878-f003:**
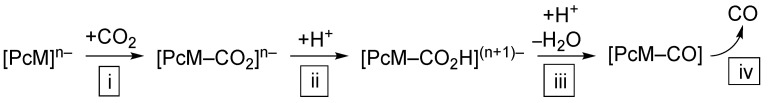
Partial reaction sequence for DFT-calculated CO_2_ reduction by PcM (M = Mn, Fe, Co, Ni, Cu).

**Figure 4 molecules-28-05878-f004:**
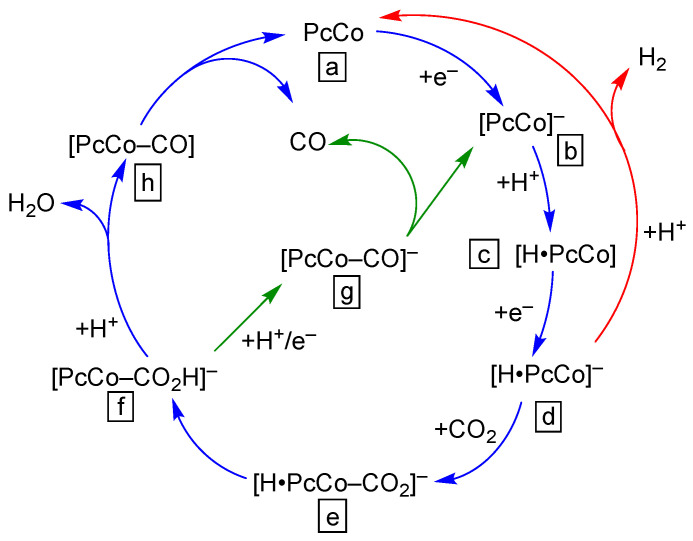
Summary of proposed mechanism(s) for PcCo-mediated reduction of CO_2_ to CO. The appropriate citations are given in the main text.

**Figure 5 molecules-28-05878-f005:**
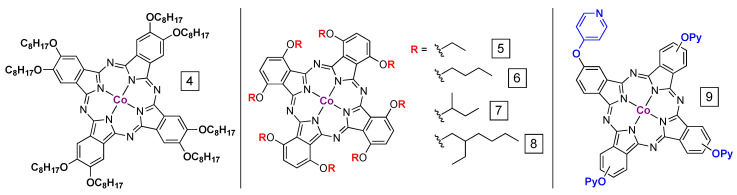
Alkoxy/phenoxy-ring substituted PcCo species used in CO_2_ reduction.

**Figure 6 molecules-28-05878-f006:**
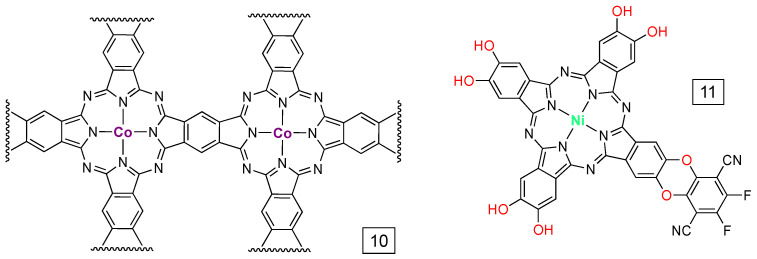
Examples of PcM complexes that have been immobilized onto carbon nanotubes (CNTs). The wavy lines indicate connections to other PcCo-derived subunits of the 2D material.

**Figure 7 molecules-28-05878-f007:**
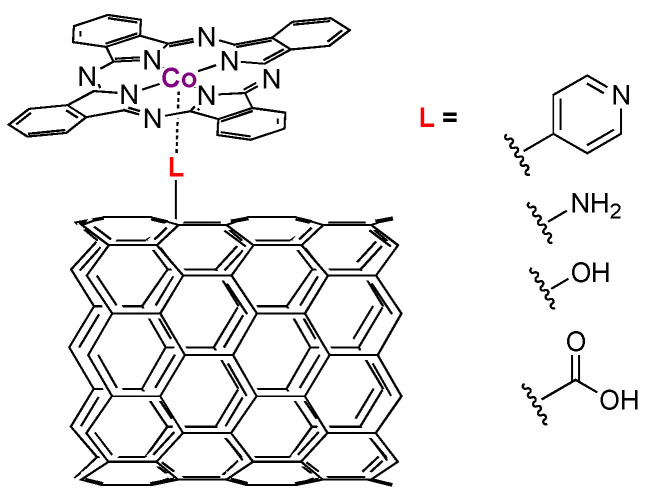
Examples of carbon nanotubes modified with PcM-ligating groups.

**Figure 8 molecules-28-05878-f008:**
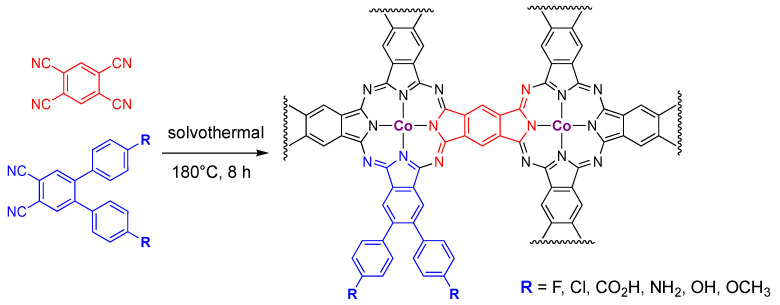
Synthetic route to PcCo materials where the physical properties of the PcCo core were modified by the identity of the R-group on the phenyl ring. The wavy lines indicated connections to other PcCo-derived subunits of the 2D material [64].

**Figure 9 molecules-28-05878-f009:**
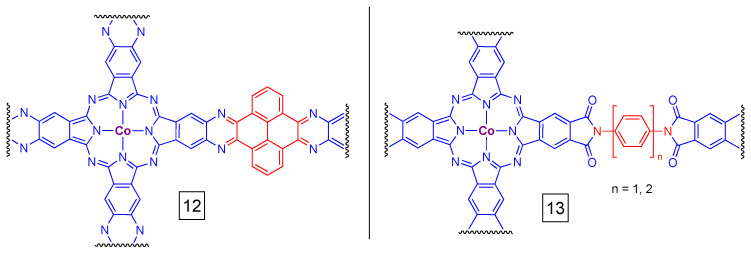
Examples of subunits of covalent organic frameworks derived from PcCo. The wavy lines indicated connections to other PcCo-derived subunits of the 2D material.

**Figure 10 molecules-28-05878-f010:**
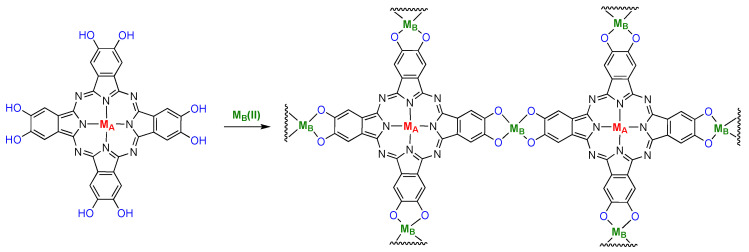
Prototype structure of conjugated metal organic framework materials comprised of metal-catecholate-linked PcM.

**Figure 11 molecules-28-05878-f011:**
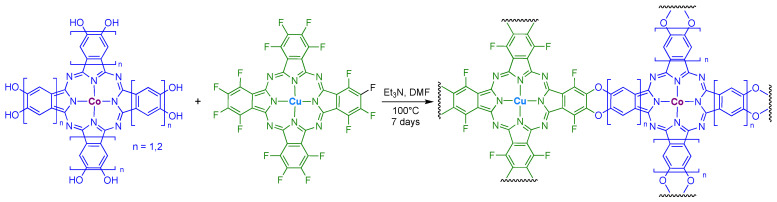
Synthetic scheme for heterobimetallic PcM dioxin-linked COFs. The *n* values (where *n* is 1 or 2) refer to the number of aromatic groups between the Pc core and the dioxin linker.

**Figure 12 molecules-28-05878-f012:**
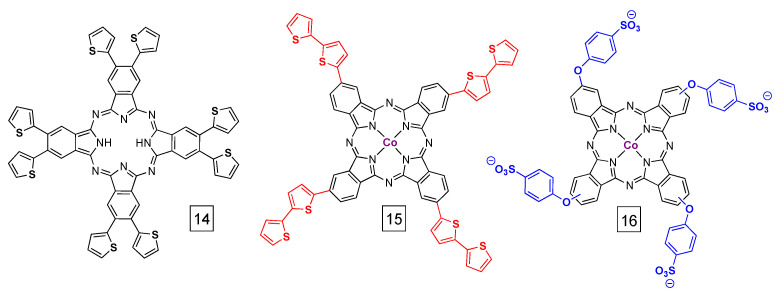
Examples of Pc-derivatives that can be electropolymerized into films (**14** and **15**) on electrodes or crosslinked with polymers (**16**).

## Data Availability

No new data was created in this Review article.

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
