# Peer review of "Contemporary Strategies for Immobilizing Metallophthalocyanines for Electrochemical Transformations of Carbon Dioxide"

_molecules, 2023, doi:10.3390/molecules28155878_

Round 1

Reviewer 1 Report

In this review article, Lawson et al. presented “a great many” examples of metallophthalocyanines and strategies to immobilize them onto solid electrodes for electrochemical CO2 reduction. The search for catalytic materials comprising molecular catalysts is current for performing fuel-forming reactions, and I believe that this review will be a good addition to the literature. Therefore, I recommend it for publication. However, I have a few suggestions that I think would improve the clarity of some of the sentences, as listed below-

1.     Page 2, lines 84-85: “The Co and Ni complexes were active for CO2 reduction...” Ref. 27 reported those catalysts “active” for CO2 reduction based on only the current enhancements in cyclic voltammograms. It was back in 1977 when scientists just started to explore this field. However, it is not always true to call a catalyst “active” for CO2RR without identifying the products. I suggest avoiding the word “active,” if there are no further publications that prove that those catalysts can actually convert CO2.

2.     Page 3, line 108: “The Faradaic efficiencies for both CO2 and CO reduction were slightly higher…” Is there any %Faradaic efficiency available in the literature? Providing some numbers would clarify the sentence further.

3.     Page 4, line 143: “The mechanisms by which different first row PcM complexes…” I think the authors mean the first row “transition metals.”

4.     Figure 3 does not look like a catalytic cycle, which the authors have described in the text. Overall, it looks like an incomplete catalytic cycle as it starts from [PcM]n– and ends with [PcM] (not shown) upon releasing CO. I would suggest completing the cycle from [PcM] to [PcM]n– by adding ne, if there is no other step exists.

5.     Page 6, line 263: “the challenges associated with the multiple interfaces involved in such systems are appreciated.” Mentioning some of the challenges could help the general audience.

6.     Page 6, line 253: A series of DFT calculations shows…” It’ll be “show.”

The language looks good to me!

Author Response

Response to Reviewer 1

In this review article, Lawson et al. presented “a great many” examples of metallophthalocyanines and strategies to immobilize them onto solid electrodes for electrochemical CO2 reduction. The search for catalytic materials comprising molecular catalysts is current for performing fuel-forming reactions, and I believe that this review will be a good addition to the literature. Therefore, I recommend it for publication. However, I have a few suggestions that I think would improve the clarity of some of the sentences, as listed below.

We thank this reviewer for their constructive comments and suggestions. Responses are listed below and marked with **

  1. Page 2, lines 84-85: “The Co and Ni complexes were active for CO2 reduction...” Ref. 27 reported those catalysts “active” for CO2 reduction based on only the current enhancements in cyclic voltammograms. It was back in 1977 when scientists just started to explore this field. However, it is not always true to call a catalyst “active” for CO2RR without identifying the products. I suggest avoiding the word “active,” if there are no further publications that prove that those catalysts can actually convert CO2.

**This is an excellent point and we revised the text to be explicit about those early observations. As suggested, we carefully reviewed our use of the word “active” to ensure that it conveys the intended meaning.

  1. Page 3, line 108: “The Faradaic efficiencies for both CO2 and CO reduction were slightly higher…” Is there any %Faradaic efficiency available in the literature? Providing some numbers would clarify the sentence further.

**The relevant Faradic efficiency values are available and are now included in the text on page 3.

  1. Page 4, line 143: “The mechanisms by which different first row PcM complexes…” I think the authors mean the first row “transition metals.”

**Correct. This change has been made on page 4.

  1. Figure 3 does not look like a catalytic cycle, which the authors have described in the text. Overall, it looks like an incomplete catalytic cycle as it starts from [PcM]n– and ends with [PcM] (not shown) upon releasing CO. I would suggest completing the cycle from [PcM] to [PcM]n– by adding ne, if there is no other step exists.

**Figure 3 was not intended to be a catalytic cycle. We added a line of text stating that the PcM complexes re-enter the proposed catalytic cycle in steps that are not shown. We also revised the caption to indicate that it is a partial series of reaction steps.

  1. Page 6, line 263: “the challenges associated with the multiple interfaces involved in such systems are appreciated.” Mentioning some of the challenges could help the general audience.

**We added the proposed information (now on page 7). The sentence reads: A partial list of such operationally important interactions includes: the interactions of catalysts with solid particulate supports, the interparticle interactions, the particle ionomer interactions, and the solvent-ionomer interactions.

  1. Page 6, line 253: A series of DFT calculations shows…” It’ll be “show.”

**This change has been made.

Reviewer 2 Report

The manuscript presents a review of recent advances in the use of metallophthalocyanines as catalysts for the electrochemical reduction of carbon dioxide. The subject is up-to-date and important, and the manuscript is well written. Therefore, in my opinion, it is worth publishing. I have only one remark. The Authors usually use the term “Faradaic efficiency” but sometimes “Faradaic efficacy”. I think it should be unified, and the term “Faradaic efficiency" is more appropriate.

Author Response

Response to Reviewer 2.

The manuscript presents a review of recent advances in the use of metallophthalocyanines as catalysts for the electrochemical reduction of carbon dioxide. The subject is up-to-date and important, and the manuscript is well written. Therefore, in my opinion, it is worth publishing. I have only one remark. The Authors usually use the term “Faradaic efficiency” but sometimes “Faradaic efficacy”. I think it should be unified, and the term “Faradaic efficiency" is more appropriate.

**We thank this reviewer for their enthusiasm for the article. We have replaced all instances of "Faradaic efficacy" with the more appropriate term "Faradaic efficiency".

Reviewer 3 Report

The authors represented a tutorial review on the contemporary strategies for immobilizing metallophthalocyanines for electrochemical transformations of carbon dioxide. Overall, this review is well written and presented and worthy of publication. To further improve the quality of this paper, I have a couple of suggestions for the authors to consider: 

1. All the figure displayed are quite similar to one another looking dull and monotonous, which are  not attractive to the readers.

2. The authors stated “Instead, we try to focus on the underlying chemical principles that improve CO2 reduction kinetics and/or overpotentials”, about which, however, I could see few. I would suggest the authors include at least one representative reaction network of electrochemical reduction of CO2 (e.g., Nanoscale, 2018, 10, 15262) and make some comments. The metallophthalocyanines could behavior similarly as the popular single-atom catalyst which can bridge heterogeneous and homogeneous catalysis.

Author Response

Response to Reviewer 3.

The authors represented a tutorial review on the contemporary strategies for immobilizing metallophthalocyanines for electrochemical transformations of carbon dioxide. Overall, this review is well written and presented and worthy of publication. To further improve the quality of this paper, I have a couple of suggestions for the authors to consider:

**We thank this reviewer for their constructive feedback and for their positivity about this review article. Responses are listed below and marked with **

  1. All the figure displayed are quite similar to one another looking dull and monotonous, which are not attractive to the readers.

**We appreciate that ChemDraw structures are not the most striking images, but they do serve to efficiently convey important chemical structures. We revised most of the Figures to include more prominent text and colors to indicate the most important part of the image. We hope that these are more engaging presentations of structures.

  1. The authors stated “Instead, we try to focus on the underlying chemical principles that improve CO2 reduction kinetics and/or overpotentials”, about which, however, I could see few. I would suggest the authors include at least one representative reaction network of electrochemical reduction of CO2 (e.g., Nanoscale, 2018, 10, 15262) and make some comments. The metallophthalocyanines could behavior similarly as the popular single-atom catalyst which can bridge heterogeneous and homogeneous catalysis.

**This point is well taken and that sentence in the introduction has been revised (on page 1-2) to read: Instead, we try to focus on some of the cross-cutting approaches to making molecules and materials that have favorable CO2 reduction kinetics and/or overpotentials. In that light we also emphasize areas where research has lagged or can be further expanded.

The suggested reference has been added on page 12. It is included in a new section that describes single-site CO2 reduction catalysts that are related to metallophthalocyanine electrocatalysts.

Reviewer 4 Report

Since metallophthalocyanine (PcM) coordination complexes are well-known mediators of electrochemical reduction of carbon dioxide (CO2) and have many properties that show promise for practical applications in the energy sector, the authors reviewed their progress in the last decade as well as several promising approaches to incorporate PcM into a range of materials, from simple carbon-adsorbed preparations to extended polymer networks. The advantages and drawbacks of these approaches have been discussed. Moreover, the authors also described developments in the immobilization of PcM CO2 reduction catalysts in the last decade (2013 to 2023) and proposed promising avenues and strategies for future research. The research is meaningful, and I would like to recommend its publication after minor revisions.

(1)    For COFs and polymers, the shape and size did have strong effect on the catalytic properties. It should be discussed in a separated paragraph.

(2)    In PcM structures, the donor or accepted functional groups would have a strong effect on the catalytic behaviors. It should be discussed in the revised manuscript.

(3)    Some in situ characterization methods to understand catalytic mechanism should be summarized in a separated paragraph.

(4)    One related reference might be included in the revised manuscript: 10.1021/acsnano.2c06059

(5)    TOC is required

Author Response

Response to Reviewer 4.

Since metallophthalocyanine (PcM) coordination complexes are well-known mediators of electrochemical reduction of carbon dioxide (CO2) and have many properties that show promise for practical applications in the energy sector, the authors reviewed their progress in the last decade as well as several promising approaches to incorporate PcM into a range of materials, from simple carbon-adsorbed preparations to extended polymer networks. The advantages and drawbacks of these approaches have been discussed. Moreover, the authors also described developments in the immobilization of PcM CO2 reduction catalysts in the last decade (2013 to 2023) and proposed promising avenues and strategies for future research. The research is meaningful, and I would like to recommend its publication after minor revisions.

**We thank this reviewer for their constructive comments and suggestions. Responses are listed below and marked with **

(1)    For COFs and polymers, the shape and size did have strong effect on the catalytic properties. It should be discussed in a separated paragraph.

**A new paragraph has been added on page 12 that gives a short overview of the importance of pore size on carbon dioxide reduction mediated by COFs.

(2)    In PcM structures, the donor or accepted functional groups would have a strong effect on the catalytic behaviors. It should be discussed in the revised manuscript.

**We make note of the importance of different functional groups

(3)    Some in situ characterization methods to understand catalytic mechanism should be summarized in a separated paragraph.

**We have added a new paragraph highlighting some of the modern in situ techniques that have informed the mechanisms outlined in section 3. The new paragraph is on page 5.

(4)    One related reference might be included in the revised manuscript: 10.1021/acsnano.2c06059

**We now include this review article with the new section about single site catalysts. We note that we do not extensively cite review articles. It is our opinion that reviews themselves are more useful in citing the original literature.

(5)    TOC is required

**A TOC graphic is include with the revised submission.